# The Potential Supply and Demand of Farmers' Land Contract Rights-Based on 697 Households in Four Provinces of China

**Wujing Wang [1] and Xingqing Ye [1,2,*]**

[1]   National Agricultural and Rural Development Research Institute, China Agricultural University, Beijing 100083, China; wangwujing@cau.edu.cn

[2]   Research Department of Rural Economy, Development Research Center of the State Council, Beijing 100010, China

[*]   Correspondence: 02025@cau.edu.cn; Tel.: +86-010- 51780015

**Abstract:** A new urbanization and rural revitalization strategy has been implemented in China over a number of years, under which farmers' land contract rights (LCRs) flow inevitably through various means. The practice in reform pilot areas indicates that government funds cannot meet all the needs, so exploring market-based LCR payout paths is important for rural land tenure system reform. The purpose of this study is to answer questions such as the following: How would farmers respond if they were allowed to trade LCRs? Is there an equilibrium point between the potential supply and demand of LCRs? Which factors would affect the potential supply and demand of LCRs? In this study, 697 valid questionnaires from Ningxia, Hebei, Henan, and Shandong provinces, China, were used for analysis by the multiple bounded discrete choice (MBDC) method and MBDC-Tobit model. The results show that there is a potential market among rural collective households in China, with an equilibrium price of ¥27,800/mu ($59,714.4/ha), and a proportion of farmers who are willing to buy or sell LCRs is around 10.0%. The factors affecting the potential supply and demand of LCRs include land grade, average agricultural income per unit, total money to buy urban houses and cars, age, number of household members with a college education or above, and risk appetite. If the institutional barriers that hinder LCR transactions were eliminated, the potential supply and demand of LCRs would be matched, and the market would provide funds for next-stage reforms.

**Keywords:** land contract rights (LCRs); land reform; land tenure; land rights; MBDC

## 1. Introduction

### 1.1. Background

The dual urban–rural land tenure system has been implemented for decades in China. Urban land is owned by the state and managed by the government. Rural land is collectively owned by farmers and is managed by rural collective economic organizations. Since the reform and opening-up, urban land has been traded for a long time, and "land finance" has helped China experience rapid economic development. However, in order to ensure a basic living for farmers, rural land transactions are prohibited. With the improvement of China's modernization, urban–rural integration has become a major trend. The land tenure reform for separating rural land ownership rights, land contract rights (LCRs), and management rights (the 3R policy) has led to the rapid development of a land leasing market and a reduction of farmers' dependence on land.

However, due to the lack of a mechanism for paying out LCRs, farmers are still prohibited from trading their LCRs legally, even if they have settled in the city and are not engaged in farming, leaving

the land uncultivated [1–5]. With the deepening development of industrialization and urbanization, Chinese households have gradually differentiated into various types, and their dependence on LCRs and preference for land tenure have clearly changed. According to the National Bureau of Statistics of China, at the end of 2018, China's urbanization rate reached 59.58%, and there was a floating population of 241 million. Among these migrants, some have settled in cities and have encouraged relatives to move to the cities. In addition, the trend gradually changed from personal migration to family migration [6]. Some farmers have been engaged in non-agricultural industries for a long time and have no livelihood dependence on LCRs. Some rural elderly, whose children have settled in cities to work, are unwilling to migrate to their children's cities because they have to maintain a basic livelihood by planting a small amount of contracted land. As these farmers grow older, they are willing to trade their legal LCRs in exchange for pensions.

At the same time, in the context of the integrated development of urban and rural areas, on the one hand, some farmers who have accumulated sufficient capital and management experience in cities suitable for starting up in rural areas have a strong willingness to obtain stable land property rights and invest in agriculture over the long term. On the other hand, some farmers who are engaged in the agriculture industry want to obtain stable property rights in order to scale up. With the differentiation of households, some farmers are willing to sell their LCRs, while others are willing to buy some LCRs, but the potential transactions are prevented by the current land tenure system.

The government has set up reform pilot areas to carry out trials, allowing farmers to sell their LCRs to the local government, and these trials have achieved some results. However, because of the limited funding, these trials cannot meet all the needs of farmers who are willing to sell their LCRs. Based on the evolution of Chinese policies, the policy of how to deal with farmers' LCRs has changed; policy is now focused on transfer rather than maintenance [7]. The 13th Five-Year Plan for National Economic and Social Development, released in 2016, asserts protection of farmers' LCRs and support for voluntary LCR payout legally in the pilot areas. In these pilot areas, most local governments fully respect the farmers' willingness to withdraw LCRs in order to practice land reform, even though the funds from the government are limited.

The local government of Liangping District, Chongqing, one of the reform pilot areas, raised ¥1 million ($143,200) for farmers' LCR payout. As of July 2017, 21 of the 70 households in Chuanxi village applied to sell their LCRs to the local government, but according to threshold conditions, only 15 households were approved, and 82.12 mu (5.47 ha) were bought by the government at a price of ¥14,000/mu ($30,072/ha). The withdrawal strategy completely alleviates the concern of farmers who are no longer engaged in agriculture and increases those household funds to settle in a city. Although it is welcomed by farmers, it is difficult for this LCR withdrawal strategy to meet the needs of all famers who are willing to sell their LCRs because of the limited funding.

Pingluo County, Ningxia Province, is another typical successful case of a national land reform pilot area. The local government bought household LCRs and then sold them for migration to alleviate the poverty at a lower cost by using national poverty alleviation funds. The "borrow ships to sail" strategy improves the efficiency of use of idle resources in rural areas. The concern is that if the national poverty alleviation funds run out, it will be difficult to continue the reform in the future. It is difficult to meet all needs by relying solely on government funds due to financial constraints.

Can we try to explore a market-based LCR withdrawal mechanism? We need to determine whether there is a potential LCR market in rural areas and whether there is an equilibrium point between the potential supply and demand of LCRs across collectives in rural China. In addition, we need to determine the characteristics of farmers who can be potential suppliers and buyers of LCRs.

Based on empirical analysis, this study scientifically answers these questions and could help the Chinese government design a better land tenure system to revitalize rural land resources on the basis of the 3R policy, which is important to deepen the reform of rural land tenure systems.

*1.2. Current Research*

According to Chinese laws and policies, households only have rights to land for cultivation during the legal contract period. In the past years, the 3R policy has given households the right to lease but not sell LCRs. In fact, the land tenure system cannot adapt to the changes in current times, which require a bidirectional flow of urban and rural elements for integrated development. Therefore, scholars in different fields have conducted a series of discussions.

Scholars have explored the possible directions of land tenure system reform and the mechanism of paying out household LCRs. Du and Ren believed that the static empowerment of household LCRs would violate justice and equality. Those who have no rural collective membership enjoy LCRs and social security repeatedly, and this goes against fairness and impartiality [8]. Zhong and Li believed that the complete withdrawal of the rural population refers to the withdrawal of not only farmers' economic relations but also legal relations, that is, the complete withdrawal of rural collective membership [9]. The complete withdrawal of households from rural areas requires the simultaneous withdrawal of the "three rights", consisting of land contract rights, the right to use house sites, and the distribution right of collective income [10–12]. Guo and Zhang supposed that the lack of an LCR payout mechanism is not conducive to either realizing the land rights of farmers settled in cities or the advancement of urbanization, and the inequality of new land rights could cause social conflicts [13]. Chamberlain's research proved that the reform of the land property rights system and the stability of farmers' property rights in South Africa contributed to the success of inclusive businesses [14].

In addition, scholars have conducted research on farmers' LCR payout mechanism. Wang believed that in order to improve the payout mechanism, farmers should ensure their freedom of choice. The focus was on establishing procedures, cooling-off periods, gravitational mechanisms, linkage support policy systems, and full compensation mechanisms for the withdrawal of rural household land management rights [15]. Guo proposed changing static collective membership to dynamic collective membership [16]. Gao and Song believed that successful reform of LCR payout should be through a joint mechanism of policy, cooperative, and market [17].

Some research mainly focuses on the willingness of farmers to sell their LCRs. Farmers who are willing to sell their LCRs at certain prices exist. Zhang's survey in Henan, Hunan, and Chongqing found that among 886 households in 3 provinces or cities, 19.64% of farmers were willing to sell their LCRs [18,19]. Liu's survey of 779 households in Shandong, Hebei, and Henan found that the proportion of farmers who were willing to sell their LCRs was 21.7% [20,21]. Gao and Li's survey of 579 households in the Guanzhong area of Shaanxi province found that 24.20% of farmers were willing to sell their LCRs [22].

Regarding the research on the factors that influence the willingness to sell LCRs, Zhang, He, and Gu believed that the differentiation of households would not affect the farmers' willingness to sell LCRs [23]. Li and Ye used empirical data to prove that family income has an important impact on the willingness to sell LCRs. The richer the farmers, the less willing they are to sell LCRs [11]. Yang, Ren, and Du believed that confirmation of land rights and compensation price have a positive impact on migrant workers' willingness to sell LCRs [24]. The influential factors of a farmer's willingness to sell LCRs also include the occupations of those working off the farm, the rate of leaving the farm, the land rental rate, stable non-agricultural income, pension, and compensation price [18]. Other factors include age, participation in new rural insurance, farmland area, labor force, property rights recognition, economic compensation [25], urban housing assets [26], household registration system reform, etc. [27].

In countries where land is private, scholars have mainly focused on the land transaction behavior of farmers. Their research results show that land transaction behavior is affected by non-agricultural employment of farmers [28], land transaction prices [29], transaction models [30], discounted value from land farming and the value of land as collateral [31], expectation of future value appreciation [32], willingness for future generations to inherit [33], urban–rural income gap [34], farmland size [35], farmer debt [36], taxation [37], and other factors.

*1.3. Aims of the Study*

Policymakers, who usually depend on a thinking path, tend to rely on previous successful experiences to formulate relevant policies today, and they are not sensitive to the needs of the new land tenure system with the changing times [38]. Faced with the current incompatibility between rural land policies and the situation of economic development, the government has only tried to resolve the contradictions by means of local government funding. However, due to insufficient funding, the land tenure system reform is limited. Research has also focused on policy-based withdrawal mechanisms and empirical evidence, which is insufficient. Therefore, the main purpose of this research is to prove whether it is feasible to explore a market-based withdrawal mechanism in rural areas of China to make up the funds to promote the reform of the land tenure system.

Specifically, through the survey, it is critical to answer questions such as the following: Are there potential suppliers and buyers of LCRs? Is there an equilibrium point between potential supply and demand? What factors affect the potential supply and demand of LCRs? If the answers are affirmative, it means that as long as farmers are allowed to sell their LCRs, some buyers will pay for them.

## 2. Materials

The distribution area of farmers who have actually completely sold their LCRs is critical to this study. At the end of 2014, the Chinese government listed the withdrawal of LCRs as one of the 14 reform test tasks in the "Official Reply on the Second Batch of Rural Reform Pilot Areas and Test Tasks". After 4 years, there are a few pilot areas for farmers to sell their LCRs voluntarily, and the number of households involved is not large. Therefore, it is difficult to choose the area to be investigated.

*2.1. Research Area Selection*

Pingluo County, Ningxia Province is the reform pilot area with the largest number of households who had been paid out in China. As of July 2018, 2036 farmers have sold part or all of their LCRs and other rights, which has accumulated some valuable experience. The sold LCRs and farmers' houses are targeted for settling poor immigrants as defined by the government, so the LCRs are collected by village collectives in accordance with membership units and paid by the local government. One membership unit is five mu (0.33 ha) LCR plus 80 m$^2$ of rural housing ownership with one to three members. Households in the research included those that sold only part of their LCRs.

Because the Yellow River passes through this area, Pingluo County is rich in water resources and has a flat terrain. The area is suitable for land concentration and large-scale farming. In addition, Ningxia's urbanization rate is 57.98%, which is close to that of Shandong's 60.58%, Hebei's 55.01%, and Henan's 50.16%, and the migrant population characteristics are also close. The experience is more suitable for promotion in the North China Plain. Therefore, the reform pilot area selected in this study was Pingluo County, Ningxia Province, and the non-reform pilot areas Hebei, Shandong, and Henan provinces were selected. The distribution was shown in Figure 1.

*2.2. Investigating in the Reform Pilot Area*

According to the farmland production, all of 13 towns in Pingluo County were divided into three categories by the local government. There were three towns in the first category, and the price of LCRs was ¥10,000/mu ($21,480/ha); there were six towns in the second category, and the price was ¥9,000/mu ($19,332); there were four towns in the third category, and the price was ¥8,000/mu ($17,184/ha). At first, stratified sampling method was adopted, that is, random sampling in each category, one town in the first category, two towns in the second category, and one town in the three categories (see Table 1). Then, the number of households who had been paid out in each town was sorted, and the top five villages were selected. If the number of households was less than five, the village was no longer selected. Geographically, the 18 villages selected are located in the south, north, central east, and central west of Pingluo County.

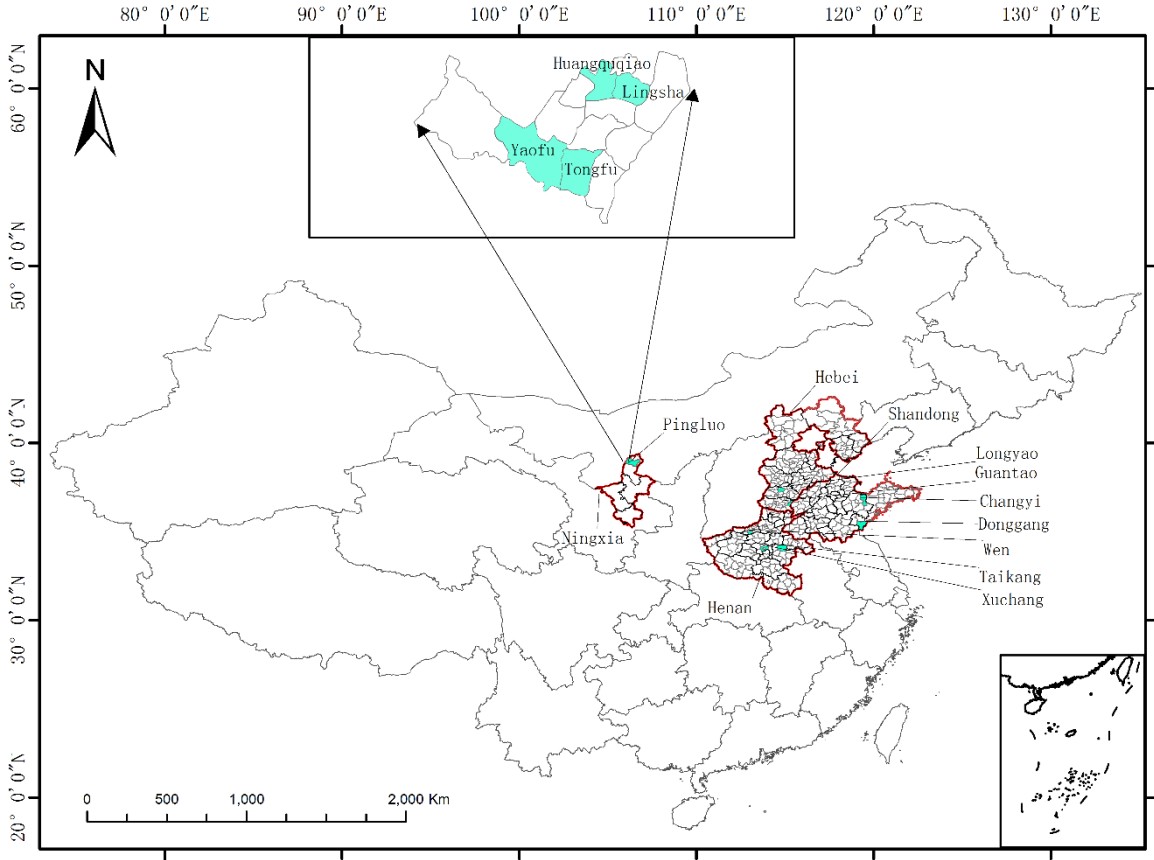

**Figure 1.** The distribution of survey areas.

**Figure 1.** The distribution of survey areas. (Data source: http://bzdt.ch.mnr.gov.cn/index.html. Map approval number is GS (2019)1823).

**Table 1.** Survey data by town in Pingluo County, Ningxia Province (the reform pilot area).

| SN | Town | Number of Households Who Had Their LCRs Paid Out [1] | Number of Households Who Did Not Have Their LCRs Paid Out [2] | Price Specified by Government (¥1000) |
|---|---|---|---|---|
| 1 | Lingsha | 54(96) | 75(75) | 9 |
| 2 | Huangquqiao | 52(86) | 67(75) | 9 |
| 3 | Yaofu | 49(78) | 57(60) | 8 |
| 4 | Tongfu | 26(82) | 56(60) | 10 |
| | Total | 181(342) | 255(270) | |

Note: (1) Figures in parentheses are total numbers of households, and figures outside parentheses are actual survey numbers. (2) Figures in parentheses are expected survey numbers, and figures outside parentheses are actual survey numbers.

At present, most of the households that received payouts have settled in cities, and it was difficult to contact them directly. Therefore, in order to understand the basic situation and obtain the contact information of the households and get in touch with them, the investigator needed the help of village leaders. The investigator then conducted the survey by telephone with their consent, with a subsidy of ¥20 ($2.86). For the households that did not sell their LCRs, 15 households were randomly selected in each village and were interviewed face-to-face with the same subsidy.

### 2.3. Investigating in the Non-Reform Pilot Area

The respondents of the non-reform pilot areas were from Shandong, Hebei, and Henan provinces, and the total number of samples was expected to be around 300. Multistage sampling method was adopted. A total of 172 counties in Hebei Province, 140 counties in Shandong Province, and 157 counties

in Henan Province were numbered; then, two counties in Hebei, two counties in Shandong, and three counties in Henan were randomly selected. If the sample distribution was too concentrated, they would be randomly selected again. Finally, the seven counties selected are relatively evenly distributed in the North China Plain (see Table 2). With the same method, two towns in each county were randomly selected, two or three villages were randomly selected in each town according to the town size, and 15 households were randomly selected in each village and interviewed face-to-face with their consent; they received a subsidy of ¥20 ($2.86).

**Table 2.** Survey data by town in three provinces (the non-reform pilot area).

| SN | Province | County (District) | Town | Number of Households [3] |
|----|----------|-------------------|------|--------------------------|
| 1 | Shandong | Rizhao Donggang | Xianghejie | 28(30) |
| 2 | | | Wolongshan | 32(30) |
| 3 | | Changyi | Weizijie | 29(30) |
| 4 | Hebei | Longyao | Lianzi | 26(30) |
| 5 | | | Longyao | 31(30) |
| 6 | | Guantao | Guantao | 34(30) |
| 7 | | | Chaibao | 16(30) |
| 8 | Henan | Wen | Zhaobao | 24(30) |
| 9 | | | Nanzhangqiang | 25(30) |
| 10 | | Xuchang Jianan | Changcunzhang | 22(30) |
| 11 | | | Lingjing | 11(15) |
| 12 | | Taikang | Maozhuang | 12(15) |
| 13 | | | Matou | 24(30) |
| | | | Total | 304(360) |

Note: (3) Figures in parentheses are expected survey numbers, and figures outside parentheses are actual survey numbers.

The data are mainly from households surveyed by the author's team from July to September 2018. In all, 740 valid questionnaires were completed, of which 43 households had no LCRs for reasons such as no LCRs during the second round of contracting, land requisitions or they were already paid out. Therefore, there were 697 valid questionnaires after removing these samples. Among them, in reform pilot areas, there were 158 households that had part of their LCRs paid out and 252 households that had not sold their LCRs, and there were 287 households in non-reform pilot areas.

## 3. Methods

### 3.1. Analytical Approach

In order to find whether there is a potential LCR market in rural China, it is critical to analyze the situation by economic behavior theory.

It is assumed that farmers are rational, and LCRs utility is mainly determined by the number of LCRs q and other items x (vectors). The choice of farmers to sell or buy LCRs is based on maximizing the utility of the family under constraints.

$$Max\ u(\boldsymbol{x}\ q) \quad s.t.\boldsymbol{xp}\ =\ y. \tag{1}$$

This is the Marshallian demand function [39] derived from utility maximization.

$$x_i\ =\ x_i(\boldsymbol{p}\ q\ y). \tag{2}$$

Equation (2) is substituted into Equation (1) to obtain the indirect utility function:

$$v(\boldsymbol{p}\ q\ y)\ =\ u[x_i(\boldsymbol{p}\ q\ y),\quad q]. \tag{3}$$

Suppose the function is differentiable for both price p and income y, and it is a monotonic non-decreasing function for the number q of LCRs. That is, the more LCRs householders hold, the greater utility they have. Suppose the number of a farmer's LCRs increased from $q_0$ to $q_1$ ($q_1 > q_0$) while the farmer's income y and the price p of other items remained the same, so the utility of the farmer increased from $u_0 = v(\boldsymbol{p}\, q_0\, y)$ to $u_1 = v(\boldsymbol{p}\, q_1\, y)$, ($u_1 > u_0$). Willingness to sell (WTS) represents the price of selling LCRs, and willingness to buy (WTB) represents the price of buying LCRs. The farmer who sells LCRs can receive an income of $(q_1 - q_0)$WTS, and the farmer who buys LCRs needs to pay $(q_1 - q_0)$WTB.

According to Miller's theory [40], the following formulas can be derived:

$$v(\boldsymbol{p}\, q_1\, y - (q_1 - q_0)WTS) = v(\boldsymbol{p}\, q_0\, y), \tag{4}$$

$$v(\boldsymbol{p}\, q_1\, y) = v(\boldsymbol{p}\, q_0\, y + (q_1 - q_0)WTB). \tag{5}$$

Equation (1) can be converted to the expenditure minimization equation:

$$\min_{\boldsymbol{x}} \sum p_i x_i \quad s.t.u = (\boldsymbol{x}\, q). \tag{6}$$

Then, a compensation demand equation can be derived: $x_i = g_i(\boldsymbol{p}\, q\, u)$, $i = 1, \ldots, N$; an expenditure equation can be derived: $e(\boldsymbol{p}\, q\, u) = \sum p_i g_i(\boldsymbol{p}\, q\, u)(q_1 - q_0)$; and WTS and WTB can be derived:

$$(q_1 - q_0)WTS = e(\boldsymbol{p}, q_0, u_0) - e(\boldsymbol{p}, q_1, u_0), \tag{7}$$

$$(q_1 - q_0)WTB = e(\boldsymbol{p}, q_0, u_1) - e(\boldsymbol{p}, q_1, u_1). \tag{8}$$

### 3.1.1. Households with Complete Elasticity of Substitution

Farmers who would completely sell their LCRs could use that income to buy real estate or other items in the city to improve their lives. The LCR q can be substituted completely with other items vector x. Assume that the LCR q can be substituted by one of the other items $x_i$ (the same conclusion can be made when other items are replaced); then, the utility function of these farmers can be expressed as:

$$u(\boldsymbol{x}\, q) = \widetilde{u}[x_1 + \varphi(q),\, x_2, \ldots, x_N], \tag{9}$$

where $\varphi(\cdot)$ is a surrogate function of the farmers' LCR q for other items, monotonically increasing, and $\widetilde{u}(\cdot)$ is a continuous, monotonically increasing, and quasi-concave function; the indirect utility function (Equation (10)) can be derived:

$$v(\boldsymbol{p}\, q\, y) = \widetilde{v}[p_1,\, p_2, \ldots, p_N, y + \varphi(q)]. \tag{10}$$

Equation (10) is substituted into Equations (7) and (8), and we obtain WTB = WTS; that is, if farmers' LCRs can be completely substituted by other items, such as urban pensions or money to buy a house or car, there is complete substitution elasticity between farmers' LCRs and other items, and this type of farmer is the main supplier in the potential LCR market. Therefore, the households with increasing elasticity of substitution are more likely to trade their LCRs.

### 3.1.2. Households with Zero Elasticity of Substitution

For some households, substitution elasticity between LCR q and another item vector x is zero. The direct utility function is

$$u(\boldsymbol{x}\, q) = \widetilde{u}\left[\min\left(q, \frac{x_1}{\alpha_1}\right), \ldots, \left(q, \frac{x_N}{\alpha_N}\right)\right], \alpha_1, \ldots,\ \alpha_N \text{ are positive numbers.} \tag{11}$$

The structure of the indirect utility function $v(\boldsymbol{p} \, q \, y)$ is complex, and it has different types in space$(\boldsymbol{p} \, q \, y)$, so we can take one of these forms for logical reasoning.

Suppose $q \leq y / \sum p_i \alpha_i$. Under this constraint, in order to maximize the direct utility function (Equation (11)), we should obtain the demand function $x_i = x_i \boldsymbol{p} \, q \, y = \alpha_i q$ and indirect utility function $u = v(\boldsymbol{p} \, q \, y) = \widetilde{u}[q, q, \ldots \ldots, q] = w(q)$. In *space* $\boldsymbol{p} \, q \, y$, farmers are not willing to either sell or buy their LCRs, which means that the marginal utility of income is zero. Now, if we suppose $q_0 \leq y / \sum p_i \alpha_i$ and $q_1 > q_0$, due to $v(\boldsymbol{p} \, q_1 \, y) > w(q_0)$ and Equations (5) and (6), it can be seen that the farmers may be willing to buy LCRs at the WTB price to make the utility value slightly larger, but the amount is not too much. To sell LCRs at the WTS price, $v(\boldsymbol{p} \, q_0 \, y + (q_1 - q_0)WTS) = v(\boldsymbol{p} \, q_1 \, y)$, that is, no matter how much monetary compensation is obtained, the utility of the farmer will not change, and it is unlikely that the farmer will sell LCRs, which means that if the elasticity of substitution between LCRs and other private goods is decreasing, the farmer may be willing to buy rather than sell some LCRs.

### 3.1.3. Summary of the Analysis

For most households, the elasticity of substitution between LCRs and other items is close to zero or complete, so most households can be divided into two categories: Type I farmers have greater elasticity, and type II farmers have less elasticity. For type I farmers, WTS is closer to WTB, and they are the main participants in potential LCR transactions. For type II farmers, WTS is much larger than WTB, and they may not participate in the LCR transaction. Therefore, there would be a potential trading market if there were many type I farmers.

### *3.2. Analytical Methods*

### 3.2.1. SD and MBDC Methods

In order to obtain more accurate LCR prices, the research adopted two methods from two perspectives. The first is the self-declared (SD) method, asking farmers questions such as: Assuming there are no institutional barriers for LCR transactions and there is a legal land bank (the assumption of the land bank is mainly designed to overcome the selection bias caused by the concept that the land cannot be bought and sold), do you choose to buy, sell or neither buy nor sell? If you choose to buy, what is the highest price (max-WTB) you can afford? If you choose to sell, what is the lowest compensation price (min-WTS)?

The second method is a specially designed multiple bounded discrete choice (MBDC). Based on the literature and the actual prices of LCRs in practice, a sequence with 18 prices was designed, from ¥3000/mu ($6,444/ha) to ¥300,000/mu ($64,440/ha), with questions such as: Assuming there are no institutional barriers for LCR transactions and there is a legal land bank and that the current price of LCRs is ¥30,000/mu ($64,440/ha), what is your choice? The choice of answers was 1 = definitely sell; 2 = might sell; 3 = neither buy nor sell; 4 = might buy; 5 = definitely buy. We used a large price range to cover the prices of LCRs in different regions in order to ensure sample diversity and that the survey was in line with actual conditions. One example is shown in Appendix A.

### 3.2.2. MBDC Method Introduction

The contingent valuation (CV) method is the most common non-market valuation method used by economists since the 1990s. At present, the CV methods described in the literature are roughly divided into open-ended question, payment card, dichotomous choice, and double-bound dichotomous choice methods. Although double-bound dichotomy was greatly improved based on the first three methods, it was not suitable for this study.

With respect to LCRs, on the one hand, for reasons such as different locations and household differentiation, the double bounds cannot cover the true pricing of different types of households in different regions, which is likely to cause bias; on the other hand, "yes" and "no" cannot accurately reflect true pricing willingness, especially when the true willingness is neutral, which makes it

impossible to capture the true pricing and could cause bias. The MBDC method overcomes the shortcomings of the above four CV methods and expands the price selection boundary. At the same time, it adopts a 5-point scale to accurately capture the true pricing in terms of willingness to sell.

### 3.2.3. Mathematical Foundation of MBDC

This method was first introduced in the field of economics by Welsh and Poe [41]. Modeling with this method is easy to understand. Suppose $X_{iL}$ is the lowest price and $X_{iU}$ is the highest price, that is, the price range is $[X_{iL}, X_{iU}]$. $F(X_i; \beta)$ is used to represent the probability distribution of prices, and $\beta$ is the parameter vector. For any price in the interval, if the respondent answers yes, then the probability of price $X_i$ is $1 - F(X_i; \beta)$. The probability of any willingness price $X_i$ falls within the specified price interval: $F(X_{iU}; \beta) - F(X_{iL}; \beta)$. The corresponding likelihood function is

$$Ln(L) = \sum_{i=1}^{n} \ln[F(X_{iU}; \beta) - F(X_{iL}; \beta)]. \tag{12}$$

When farmers give positive answers to all prices, $X_{iU} = \infty$, and when farmers give negative answers to all prices, $X_{iL} = -\infty$. Equation (12) is a general form of the discrete variable selection probability likelihood function, and it is also completely applicable to the MBDC method. If a farmer chooses "definitely sell" at a certain price, that means the farmer's pricing falls in the interval between the price and a higher price.

### 3.2.4. MBDC-Tobit Model

This model is a sequence of 18 prices by MBDC with typical truncated data characteristics, so it needs to be estimated using the Tobit model. This research refers to the MBDC-Tobit model designed by Cho et al. [42] as follows:

$$\begin{aligned} WTS_i &= X_i\beta + u_i \ X_i\beta + u_i > 0, \\ WTS_i &= 0 \ X_i\beta + u_i \le 0, \end{aligned} \tag{13}$$

$$\begin{aligned} WTB_i &= X_i\beta + u_i \ X_i\beta + u_i > 0, \\ WTB_i &= 0 \ X_i\beta + u_i \le 0, \end{aligned} \tag{14}$$

where $WTS_i$ is the price of farmer i's willingness to sell, $WTB_i$ is the price of farmer i's willingness to buy, $X_i$ is an independent explanatory variable that affects the price selection, $\beta$ is the correlation coefficient vector, and $u_i$ is the random error term.

### 3.3. Analytical variables

### 3.3.1. Dependent Variable Selection

The indicators of the dependent explanatory variables used in this research are SD-min-WTS and MBDC-min-WTS of farmers who potentially supply LCRs, and SD-max-WTB and MBDC-max-WTB of farmers who potentially demand LCRs. These four indicators are unique and can be used as dependent variables of the model.

### 3.3.2. Independent Variable Selection

LCR value vary according to differentiated households, including expected agricultural income, household endowment, and location attribute [43]. They should be considered when analyzing factors that influence WTB and WTS, so we mainly choose the factors that have been summarized in literatures.

1.    Factors related to the value attributes of LCRs.

Adam Smith [44] pointed out that the highest use value of an item to its owner is an important part of its price. LCR has economic resource value and farmers could benefit by farming or getting

rent. The value attributes of LCRs can be represented by land quality perfectly, but the indicator is usually assessed by soil depth, fertility, slope, exposition to sun, climate conditions, topography, etc. It is impossible to get valid data from famers. Therefore, the variable land grade was selected over land quality.

Land grade is defined as: Low-grade land can be cultivated for one season with poor irrigation, medium-grade land can be cultivated for one season with good irrigation, and high-grade land can be cultivated for two seasons with good irrigation. The variable income per unit is average agricultural income per unit, which is total agricultural income of a family divided by farm size. Farm size is the total farmland areas.

In addition, the value attributes of LCRs can be represented by land rent, but land rent can be replaced by agricultural income per unit, and it could lead to collinearity in the econometric model. Therefore, Lease duration of LCR was selected as instrumentation variable.

2. Factors related to rural household characteristics.

The characteristics of the household could be mainly reflected by family population and family financial situation [5]. The number of household members and composition of the household could affect the willingness to sell LCRs [31,45]. The variable family size means there are how many members in a household.

Variables related to family composition are generally described if there are household members settled in cities [20]. It was found that variables related to urban–rural, such as education level of household members, pension of household members, affected WTS or WTB [31,46]. The variable per_city is the proportion of urban settlers in a household. The variable num_college is the number of household members who have received a college education or above. The variable num_pension is the number of household members who have received urban pensions. Therefore, these indicators were selected.

Household asset status could reflect household income [47,48]. Per capita income is generally used to evaluate household financial situation [5]. However, the structure of a household income sources is more complex, and because it involves privacy, it is more difficult to obtain data directly. On the other hand, farmers could clearly tell the investigators whether they bought urban houses and cars and how much they spent, which is easy to obtain accurately.

3. Factors related to characteristics of heads of households.

It is generally believed that age is an important factor that significantly affects farmers' willingness to sell LCRs to settle in cities [49–53]. The education level of household heads is regarded as an important factor that affects farmers' willingness to sell LCRs [51,53,54]. The variable education level is the total years that household heads go to school. Risk appetite is also an important indicator that may affect WTS and WTB [55]. Therefore, these indicators were selected.

The measurement of risk appetite in this paper referred to the questionnaire designed by Li [56]. The specific choices were as follows. Your investment preference is 1 = usually deposit in banks and cannot afford the principal loss, 2 = willing to take small risks and make short-term investments, 3 = willing to take appropriate risks and make long-term investments, 4 = actively seeking profitable products in order to pursue better returns and willing to take risks that match the returns. In order to obtain accurate data, three questions were previously designed with verifications: Does your family buy wealth management products with higher yields than banks? If so, what is the rate of return? What is the percentage of investment in your household funds?

4. Factors related to the attributes of location.

Economic location may affect WTB and WTS [57]. We used the variable the gross domestic product (GDP) index instead of the economic development index of the farmer's city as the independent variable.

Assuming the GDP index of Shizuishan City, Ningxia, is 1 as the GDP in 2018 was ¥53.50 billion ($7.66 billion). The value of GDP/¥53.50 billion in the city where farmers are located is used as the city's GDP index. The GDP index of those cities are Shizuishan City, Ningxia Province, =1.000; Xingtai City, Hebei Province, =4.020; Weifang City, Shandong Province, =11.974; Rizhao City, Shandong Province, =4.241; Xuchang City, Henan Province, =5.291; Jiaozuo City, Henan Province, =4.433; Zhoukou City, Henan Province, =5.023.

## 4. Results

### 4.1. Descriptive Statistics of Variables

#### 4.1.1. Dependent Variable

Table 3 shows that 403 farmers were willing to sell LCRs. The average SD-min-WTS is ¥73,300/mu ($157,448.4/ha), the average MBDC-min-WTS is ¥76,270/mu ($163,827.96/ha), the average SD-max-WTB is ¥51,690/mu ($111,030.12/ha), and the average MBDC-max-WTB is ¥49,680/mu ($106,712.64/ha). It can be seen that the average price in terms of farmers' willingness to sell LCRs is higher than the average price in terms of farmers' willingness to buy.

**Table 3.** Definitions and statistical descriptions of main variables. SD, self-declared; WTS, willingness to sell; WTB, willingness to buy; MBDC, multiple bounded discrete choice.

| Variable | Explanation | Obs | Mean | Std Dev | Min | Max |
|---|---|---|---|---|---|---|
| SD-Min-WTS | Farmers' willingness to sell at the lowest price measured by the SD method (¥10,000/mu)/($10,000/ha) | 403 | 7.33/ 15.75 | 7.47/ 16.04 | 0.8/ 1.72 | 50/ 107.4 |
| SD-Max-WTB | Farmers' willingness to buy at the highest price measured by the SD method (¥10,000/mu)/($10,000/ha) | 188 | 5.17/ 11.10 | 6.76/ 14.52 | 0.1/ 0.22 | 30/ 64.44 |
| MBDC-Min-WTS | Farmers' willingness to sell at the lowest price measured by the MBDC method (¥10,000/mu)/($10,000/ha) | 403 | 7.63/ 16.38 | 7.72/ 16.58 | 0.8/ 1.72 | 30/ 64.44 |
| MBDC-Max-WTB | Farmers' willingness to buy at the highest price measured by the MBDC method (¥10,000/mu)/($10,000/ha) | 188 | 4.97/ 10.67 | 6.40/ 13.75 | 0.3/ 0.64 | 30/ 64.44 |
| age | Age (years) | 697 | 51.39 | 11.24 | 24 | 80 |
| education | Education level (years) | 697 | 7.29 | 3.65 | 0 | 19 |
| family size | Family size (persons) | 697 | 5.07 | 1.91 | 1 | 16 |
| num_college | Number of members with college education or above (persons) | 697 | 0.58 | 0.86 | 0 | 5 |
| per_city | Members settling in cities and towns (%) | 697 | 27.14 | 37.14 | 0 | 100 |
| num_pension | Members receiving an urban pension (persons) | 697 | 0.40 | 0.98 | 0 | 8 |
| risk_score | Risk appetite (1–4) | 697 | 1.63 | 1.04 | 1 | 4 |
| year_rent | Lease duration of contracted land (years) * | 697 | 1.42 | 2.89 | 0 | 20 |
| farmland size | Land size (mu)/(ha) | 697 | 13.29 /0.89 | 18.01 /1.20 | 0/0 | 355/ 23.67 |
| income_per unit | Average agricultural income per unit (¥10,000/mu/year)/($10,000/ha/year) | 697 | 0.09 /0.19 | 0.09 /0.18 | 0/0 | 0.48 /1.03 |
| land grade | Land grade (1 = low; 2 = medium; 3 = high) | 697 | 1.54 | 0.79 | 1 | 3 |
| asset_home | Average total amount to buy houses and cars (¥10,000)/($10,000) | 697 | 18.65 /2.67 | 29.87 /4.28 | 0/0 | 405 /58 |
| index_dev | Regional economic development index | 697 | 2.90 | 2.75 | 1 | 11.97 |

\* A total of 224 of the 697 households sampled leased out contracted land, with specific lease periods, and households that did not lease land were considered to be leased for 0 years.

#### 4.1.2. Variables Related to the value attributes of LCRs

The "land grade" index adopted in this paper is mainly divided into three types according to climate and irrigation conditions, with an average value of 1.541 (see Table 3). The average agricultural income per unit (total agricultural income divided by the scale of land) is ¥890/mu/year ($1,911.72/ha/year) (see Table 3). The scale of the land is the total contracted land area minus the area that has been sold or leased out, plus the area of leased land. One household has 18.0 mu (1.2 ha) of land on average (see Table 3). The average lease duration of contracted land is 1.42 years (see Table 3).

#### 4.1.3. Variables Related to Rural Household Characteristics

Table 3 shows that among the 697 households surveyed, there are 5.07 persons per household on average. The ordinary family in rural areas includes a young couple and one or two children plus the husband's parents, which accords with the structural and cultural characteristics of Chinese rural families. Many rural families do not have members with a college education or above. An average

of 0.58 people in each household have received a college education or above. On average, 27.14% of members in each household settle in cities and towns. The average total amount to buy houses and cars is ¥186,450 ($26,700).

### 4.1.4. Variables Related to Characteristics of Heads of Households

Table 3 shows that the average age of the head of a household is 51.383 years; that is, the land management decisions of rural families in China mainly rely on older farmers, which is in line with the actual situation in rural China. The average education level of household heads is 7.29 years, and the risk appetite is 1.63 on average.

### 4.1.5. Variables Related to the Attributes of Location

Table 3 shows that the average regional economic development index of selected areas in Ningxia, Hebei, Henan, and Shandong is 2.90.

### *4.2. Equilibrium Points of Potential Supply and Demand of LCRs*

Table 4 shows the statistics of the selection frequency of 18 prices by 697 farmers (MBDC method). As the assumed price of LCRs gradually increased from ¥3000/mu ($6444/ha) to ¥300,000/mu ($64,440/ha), the proportion of farmers who chose to "definitely sell" increased from 0% to 50.93%, the proportion who chose "might sell" increased from 0% to 9.33%, the proportion who chose "might buy" decreased from 1% to 0.72%, the proportion who chose "definitely buy" gradually decreased from 31.71% to 0.29%, and the proportion who chose "neither buy nor sell" decreased from 67.29% to 39.45%.

**Table 4.** Statistics on 697 households' selection frequency. LCRs, land contract rights.

| SN | Price of LCRs (¥10,000/mu)/($10,000/ha) | Definitely Sell (%) | Might Sell (%) | Neither Buy nor Sell (%) | Might Buy (%) | Definitely Buy (%) |
|---|---|---|---|---|---|---|
| 1 | 0.3/0.64 | 0 | 0 | 67.29 | 1.00 | 31.71 |
| 2 | 0.4/0.86 | 0 | 0 | 67.58 | 1.15 | 31.28 |
| 3 | 0.5/1.07 | 0 | 0 | 67.86 | 1.87 | 30.27 |
| 4 | 0.6/1.29 | 0.14 | 0 | 69.01 | 2.73 | 28.12 |
| 5 | 0.8/1.72 | 0.57 | 0.43 | 70.16 | 4.16 | 24.68 |
| 6 | 1/2.15 | 1.72 | 3.16 | 71.16 | 5.74 | 18.22 |
| 7 | 1.3/2.79 | 3.73 | 6.60 | 71.31 | 4.16 | 14.20 |
| 8 | 2/4.30 | 9.90 | 10.76 | 63.85 | 3.59 | 11.91 |
| 9 | 3/6.44 | 17.79 | 9.76 | 60.55 | 2.44 | 9.47 |
| 10 | 4/8.59 | 22.81 | 8.75 | 58.11 | 3.30 | 7.03 |
| 11 | 5/10.74 | 25.82 | 9.61 | 55.95 | 2.30 | 6.31 |
| 12 | 6/12.89 | 29.27 | 8.32 | 54.95 | 1.72 | 5.74 |
| 13 | 7.5/16.11 | 31.28 | 9.90 | 51.36 | 1.87 | 5.60 |
| 14 | 10/21.48 | 39.74 | 10.04 | 43.62 | 3.16 | 3.44 |
| 15 | 12.5/26.85 | 43.19 | 6.89 | 44.19 | 4.02 | 1.72 |
| 16 | 15/32.22 | 44.91 | 6.60 | 44.48 | 3.59 | 0.43 |
| 17 | 20/42.96 | 47.49 | 8.18 | 43.33 | 0.72 | 0.29 |
| 18 | 30/64.44 | 50.93 | 9.33 | 39.45 | 0 | 0.29 |

Note: The design of the 18 non-equidistant prices is mainly considered to cover different types of land valuation in surveyed areas in order to accurately obtain the lowest compensation price (min-WTS) and the highest price (max-WTB).

According to the classification, those who choose "neither buy nor sell" could be classified to type II, and the other respondents could be classified to type I.

By adding the numbers of farmers who chose "definitely sell" and "might sell" in Table 4, we can obtain the proportion of farmers who are willing to sell LCRs at different prices, which is the potential supply curve of LCRs. By adding the number of farmers who chose "definitely buy" and "might buy" in Table 4, we can obtain the proportion of farmers who are willing to buy LCRs at different prices, which is the potential demand curve of LCRs. Plotting the two sets of data, we obtain the graph in Figure 2, which shows that there is an intersection between the potential supply and demand curves of LCRs; that is, there is a potential equilibrium point in LCR transactions.

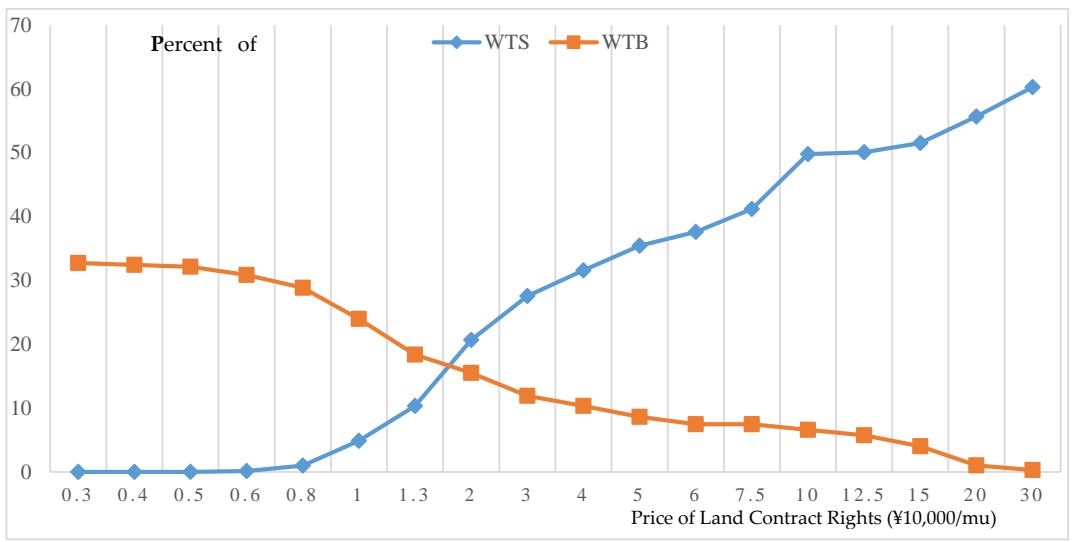

**Figure 2.** Potential equilibrium point of LCRs.

Assume that the supply or demand equation is

$$Y = aX^b, \tag{15}$$

where Y is a percentage distribution of farmers with different choices, *X* is the price of LCRs, and a and b are coefficients.

According to the above data, by fitting the curves of potential supply and demand and calculating the equations of the equilibrium point, the supply and demand equations can be estimated:

$$Y_s = 2.986X^{1.185}, \tag{16}$$

$$Y_b = 22.039X^{-0.783}, \tag{17}$$

that is, the elasticity of supply of LCRs is 1.185, and the elasticity of demand is −0.783.

An equilibrium solution can be calculated if there is an intersection on the supply-demand curve of LCRs. The equilibrium price is ¥27,800/mu ($59,714.4/ha), and the equilibrium percentage is 10.0%, that is, 10.0% of farmers are willing to buy or sell LCRs at the price of ¥27,800/mu ($59,714.4/ha), which means there is a potential LCR market among households.

### 4.3. Influential Factors of Potential Supply and Demand of LCRs

Based on the survey data, an empirical test was conducted on the factors affecting the potential supply and demand of LCRs using Stata 14.0.[1] The average variance inflation factor (VIF) between independent variables was 1.51, which shows that the collinearity among the independent variables of the model was low. The independent variables were selected according to relevant theories and studies so as to avoid omitting important variables and endogeneity. At the same time, variables were designed specifically to minimize the measurement error. Table 5 shows the regression results of the relevant models, which indicates that the overall fitting effect of the four models is good according to the log-likelihood ratio and the $R^2$ value. Models 1 and 2 are relatively robust to the results of models 3 and 4.

---

[1] StataCrop LP. 4905 Lakeway Drive College Station, TX77845 USA, Revision 22 Apr 2015.

**Table 5.** Analysis of influential factors of potential transactions of farmers' LCRs. OLS, Ordinary Least Squares.

| | Tobit (MBDC-Min-WTS) (1) | | Tobit (MBDC-Max-WTB) (2) | | OLS (SD-Min-WTS) (3) | | OLS (SD-Max-WTB) (4) | |
|---|---|---|---|---|---|---|---|---|
| | Coefficient | P-Value | Coefficient | P-Value | Coefficient | P-Value | Coefficient | P-Value |
| age | −0.05 ** | 0.01 | 0.02 | 0.44 | −0.06 *** | 0.01 | 0.01 | 0.68 |
| education | 0.03 | 0.65 | 0.10 | 0.11 | 0.01 | 0.85 | 0.10 | 0.16 |
| family size | 0.33 ** | 0.01 | 0.07 | 0.57 | 0.31 *** | 0.01 | 0.09 | 0.52 |
| num_college | −0.51 * | 0.06 | −0.50 * | 0.08 | −0.48 * | 0.08 | −0.52 | 0.11 |
| per_city | -0.01 | 0.34 | −0.00 | 0.84 | −0.01 | 0.36 | −0.00 | 0.85 |
| num_pension | 0.07 | 0.76 | −0.16 | 0.56 | 0.15 | 0.54 | −0.23 | 0.46 |
| risk_score | 0.88 *** | 0.00 | 1.55 *** | 0.00 | 0.85 *** | 0.00 | 1.66 *** | 0.00 |
| year_rent | 0.11 | 0.20 | −0.03 | 0.81 | 0.10 | 0.21 | −0.03 | 0.84 |
| farmland size | −0.02 | 0.15 | −0.03 * | 0.06 | −0.01 | 0.17 | −0.03 | 0.07 |
| income_per unit | 10.61 *** | 0.00 | 16.13 *** | 0.00 | 9.51 *** | 0.00 | 17.33 *** | 0.00 |
| land grade | 6.16 *** | 0.00 | 4.17 *** | 0.00 | 5.91 *** | 0.00 | 4.30 *** | 0.00 |
| asset_home | −0.02 ** | 0.02 | 0.01 | 0.34 | −0.02 ** | 0.02 | 0.01 | 0.42 |
| index_dev | 0.18 | 0.11 | −0.11 | 0.23 | 0.17 | 0.11 | −0.12 | 0.25 |
| _cons | −4.83 *** | 0.00 | −7.54 *** | 0.00 | −3.93 *** | 0.02 | −7.62 *** | 0.00 |
| log likelihood or $R^2$ | −1142.62 | | −430.27 | | 0.67 | | 0.84 | |

Note: * $P < 0.1$; ** $P < 0.05$; *** $P < 0.01$.

As can be seen from Table 5, the variables family size, risk appetite, average agricultural income per unit, and land grade have a significant positive effect on the min-WTS of LCRs. The variables age, college education or above, and average total amount to buy houses and cars have a significant negative effect on the min-WTS of LCRs. Other variables have no significant effects on the min-WTS of LCRs. The variables risk appetite, average agricultural income per unit, and land grade have a significant positive effect on the max-WTB of LCRs. The variables college education or above and land size have a significant negative effect on the max-WTB of LCRs. Other variables have no significant effects on the max-WTB of LCRs.

## 5. Discussion

### 5.1. Discussion of Equilibrium Point

An equilibrium point means the existence of a potential market for LCRs, and the proportion of farmers willing to sell LCRs is equal to the proportion willing to buy. If there were no institutional barriers, transactions would be conducted among collectives. It is worth nothing that it is not predicted that the potential transaction price of LCRs is ¥27,800/mu ($59,714.4/ha). It means at the equilibrium point, the LCRs are optimally allocated among type I farmers.

According to Cheung's [58] explanation of market prices, the highest price that buyers are willing to pay for LCRs is the marginal use value of LCRs. If the marginal use value is higher than the marginal market value, farmers will buy more, and if it is lower than the market value (exchange value), the farmers will not buy. This is how the farmers maximize their own utility. Therefore, when the market price is equal to the marginal value, that is, at the equilibrium price, the resources are optimally allocated. That is, there is a potential market that can solve the shortage of funds required to LCRs transaction.

### 5.2. Discussion of Potential LCRs Supploiers and Buyers

#### 5.2.1. Potential LCRs Suppliers

This study found that, many type I farmers, whether in reform pilot areas with LCRs trading experience or in non-reform pilot areas with no trading experience, had a strong willingness to sell LCRs, but due to lack of mechanisms or insufficient funds, it was difficult to meet their needs. For example, the survey in Pingluo County found that 77 out of 158 (48.74%) households that had part of LCRs paid out were willing to continue to sell their remaining LCRs, but limited poverty alleviation funds could not meet their needs. The results of a large number of other surveys in China also indicated that many people were willing to sell their LCRs (see Table 6).

Table 6. The ratio of households willing to sell LCRs.

| Author and Published Time | Survey Areas | The Ratio of Households Willing to Sell LCRs | Total Samples |
|---|---|---|---|
| Zhang Xuemin (2013) [18] | Henan, Hunan, Chongqing provinces | 19.14% | 886 |
| Liu Tongshan (2015) [20] | Shandong, Hebei, Henan provinces | 21.70% | 779 |
| Gao Jia, Li Shiping (2015) [25] | Shanxi Provicne (Guanzhong city) | 24.20% | 580 |
| Gu Haiying, Wang Changwei (2016) [26] | Shanghai, Zhejiang, Jiangsu provinces | 34.85% | 1208 |
| Li Rongyao, Ye Xingqing (2019) [11] | Chongqing (Liangping County), Sichuan (Wen, Lu counties) provinces | 63.55% | 716 |
| Liu Tongshan, Kongxiangzhi (2019) [59] | Noth China plain (6 provinces) | 44.70% | 998 |

### 5.2.2. Potential LCRs buyers

We found that there were potential LCRs buyers at different prices. The potential buyers were those who could benefit from improving the efficiency of agricultural operations. An investigation of 50 new-type professional farmers conducted by Yang [60] in Guangdong and Jiangxi provinces showed that the efficiency of agricultural technology of new-type professional farmers was increasing, which promoted the reuse of abandoned land.

Even though it was not easy to find LCRs buyers in the survey, there were some cases in reform pilot areas. One professional farmer bought out the 13-year land management right of 360 mu (24 ha) in Qinglong Village, Jiaolong Town, Liangping District, Chongqing Province for breeding goat at ¥400/mu/year ($859.2/ha/year). Another professional farmer bought 50-year LCRs of 15 mu (1 ha) in Liangping District, Chongqing Province to fish (see Appendix B).

### 5.3. Discussion of Influential Factors of Potential Supply and Demand of LCRs

### 5.3.1. Influential Factors of Potential LCRs Supply

Land grade and average agricultural income per unit have significant positive effect on farmers' min-WTS. The result is similar to some literature conclusions. William [41] pointed out that agricultural income and expected return on agricultural investment are the key factors affecting land prices. Study by Sivalai et al. [42] showed that farmer could benefit more by growing, so they value their LCRs higher.

Family size has a significant positive effect on farmers' min-WTS; that is, the larger the family size, the higher the min-WTS. This result is justified by Huang [61], against the Chinese traditional cultural background, household heads must consider the utility of three generations to ensure the safety of their lives when making decisions, so the more resources needed for survival and development, the higher the min-WTS.

The total amount to buy houses and cars has a significant negative effect on farmers' min-WTS. More money to buy houses and cars means households are richer and have more freedom of asset allocation. Under the condition that the appreciation of rural assets is limited, richer farmers might reallocate their resources to higher value-added areas, such as urban real estate. This is similar to some literature conclusions. Studies by Sivalai et al. [42] showed that given the opportunity, wealthier farmers would sell their LCRs at lower prices to withdraw from rural agricultural areas completely. Liu's research [5] showed that the households with urbanization competence were willing to sell LCRs at higher prices.

The number of household members with a college education or above has a significant negative effect on farmers' min-WTS; that is, the more members with a college education or above in a household, the lower the min-WTS. Such farmers have stable incomes and are unwilling to engage in agricultural production, so they are willing to sell their LCRs at lower prices. The result is similar to some literature conclusions. Bai et al [51] found that education level of household heads was significantly positively related to farmers' willingness to sell LCRs. Studies by Liu and Su [54] also showed that migrant women with higher education levels were more willing to give up rural land property rights.

Age has a significant negative effect on farmers' min-WTS, that is, the older household heads, the lower the min-WTS. Some senior farmers could be type I farmers. This is consistent with the results

of Wang et al. [49], Wu et al. [50] and Bai [51], whose studies indicated that the older of the respondents, the more willing to give up rural land to become urban residents.

The variables the proportion of urban-settling household members and the number of urban pensioners have no significant effect on farmers' min-WTS. These results can be justified by Scott's [62] and Ray's theory [31]. Scott's moral economics [62] stated that farmers ensure the entire family's safety according to the "safety first" principle when making decisions. Ray [31] pointed out that in order to make family life better, the internal resource allocation of the family has an altruistic effect. In general, the elderly often self-discriminate; under the premise of ensuring that the young can survive, they make decisions in their favor. Although the children of some families have settled in cities, the parents do not think that their children's jobs are stable, and they want more insurance, so it is difficult to make their mind to sell LCRs. Even if the parents obtain urban pensions, they would still retain the LCRs for their children.

### 5.3.2. Influential Factors of Potential LCRs Demand

Land grade and average agricultural income per unit have significant positive effect on farmers' max-WTB. The result is supported by some literatures, and expected return on agricultural investment is the key factors affecting land prices [5].

The variable the number of household members with a college education or above has a significant negative effect on farmers' max-WTB. This result is justified by Bai [51] and Xu [52]. The more household members with a college education or above, the more members that have stable non-agricultural jobs and urban pensions. The family's income is stable, and the risk is relatively small. With respect to the precautionary motive, the family has a lower probability of engaging in the agricultural industry and is unwilling to buy LCRs at a high price.

Risk appetite has a significant positive effect on farmers' Max-WTB; that is, household heads with higher risk appetite are more willing to buy LCRs at higher prices. This is consistent with the study of Okada et al. [44], according to which the risk appetite of major decision makers has an important impact on the willingness to pay. The more risk appetite, the higher the willingness to pay. In addition, Studies by Sivalai et al. [42] showed that risk appetite had a significant effect on the willingness to buy LCRs, and household heads with higher risk appetite are willing to pay for LCRs at higher prices. Friend and Blume [63] found that personal wealth has a positive correlation with risk appetite. Higher risk appetite means that potential buyers have more wealth; that is, some wealthy farmers are willing to participate in LCR transaction.

### 5.4. Cases of LCR Transaction across Collectives

LCR transactions in rural China are strictly restricted in non-reform pilot areas, and farmers can lease their LCRs, but they cannot sell their LCRs due to the high homogeneity of collective farmers in a small area, so it is difficult for farmers with the talent to improve land efficiency. Therefore, fewer farmers require LCRs, which makes it difficult to conduct transactions. Among collectives in a bigger area, if the institutional barriers for LCR transactions were eliminated, there would be some effective demand, which could help match the supply and demand of LCRs.

There is a real case of such a successful transaction in Liangping District, Chongqing Province, one of the reform pilot areas. (see Appendix B). The three parties, including the suppliers, the buyer, and the local government all benefited from the transaction.

### 5.5. Contribution of Research and Further Research Prospects

Academic research on potential transactions of LCRs mainly focuses on the supply side, focusing on farmers' willingness to sell and influential factors. There are several problems with existing research. On the one hand, it is easy to capture biased information by studying farmers' willingness to sell without considering price, which makes the conclusions biased; on the other hand, scholars study the potential supply and influential factors without studying the potential demand, which makes the conclusions difficult to sustain in practice.

The willingness for farmers to sell the LCR at a certain price can be realized, which needs corresponding policies and mechanisms to ensure that potential farmers can reorganize land resources, to introduce modern factors, and to create a fundamentally higher market value. For this reason, the contributions of this paper are as follows:

First, the potential supply and demand of LCRs are studied simultaneously, and reasons why a market-based path is an important direction for China's rural land system reform in the future are explored theoretically. Second, the MBDC method is used to measure the potential supply and demand price of LCRs, which makes the research more accurate and objective. Third, data from China's rural land reform pilot areas were developed and used.

Based on the research, scholars need to focus on designing a mechanism to reform rural LCRs under the framework of rural collective land ownership. The 3R policy of the rural land system allows farmers to rent across regions and collectives, but in recent years, some farmers who leased land to engage in agriculture found it difficult to make a profit, so they did not pay land rents, causing a lot of societal contradiction; that is, unstable property rights bring unstable expectations to farmers who rent and lease land, which is not conducive to the improvement of agricultural efficiency. Therefore, based on the 3R policy, reforms should be continued to allow some farmers to completely sell their LCRs to village collectives. The land will be managed and operated or leased or transferred to other farmers through the village collectives to reduce transaction costs and stabilize property rights. During the design of this mechanism, these are key questions for further research: What are the behavior characteristics of farmers? How do policymakers carry out policy design based on an understanding of the laws of farmers' behavior and social operation?

## 6. Conclusions

On the basis of the 3R policy, reform of the rural land tenure system should be continued. For those differentiated households, the government should consider eliminating institutional barriers to potential transactions, exploring marketization paths, empowering farmers who have settled in the city and obtained urban pensions with the right to LCR payout, providing options for some farmers to alleviate their concerns about land, and providing opportunities for farmers who are willing to engage in agricultural production and improve agricultural production efficiency to obtain stable land property rights across regions and collectives. This promotes the matching of supply and demand, enables farmers who are willing to withdraw from agricultural production to obtain satisfactory compensation, and enables farmers who are willing to engage in agriculture to obtain stable LCRs.

The era of China's urban–rural integration requires that Type I farmers should be given LCR trading rights. Zhou [64] believed that the current rural China has long been involved in the tide of industrialization, urbanization, and globalization. All elements are constantly flowing, transferring, and restructuring. The reform of China's rural land system cannot stop at the 3R policy, and LCRs have to be further reformed.

In addition, in the context of the rural revitalization strategy, a new type of urbanization centered on people with integrated urban–rural thinking should be promoted. Land per capita is only 1.5 mu (0.1 ha) in China, which determines that the number of rural farmers who are really engaged in agricultural production cannot be too large. The empirical results also show that farmers with urbanization ability have a strong willingness to sell LCRs. The constraints of the long-term dual urban–rural system have prevented migrant workers from enjoying equal employment and social benefits awarded to urban residents, which means most migrant workers who have worked hard in cities for a long time cannot feel a sense of urban belonging. As a result, these migrant workers place higher expectations on LCRs, which hinders the efficient allocation of rural land resources.

Therefore, new human-centered urbanization based on the integrated urban–rural thinking should continue to be promoted, and the obstacles of the dual urban–rural system should be further eliminated. All citizens of the state, regardless of where they live, should be able to enjoy urban public infrastructure and services and the efficiency, equity, and welfare of the city, which would let farmers reduce their

reliance on rural land-related rights, allow farmers who are not willing to engage in agriculture to withdraw completely from the agricultural industry, concentrate land resources with efficient farmers, improve the efficiency of rural land resources, and promote a thriving agricultural industry.

**Author Contributions:** Conceptualization, X.Y.; Methodology, W.W.; Software, W.W; Validation, W.W.; Formal Analysis, X.Y.; Investigation, W.W.; Resources, W.W.; Data Curation, W.W.; Writing-Original Draft Preparation, W.W.; Writing-Review & Editing, W.W.; Visualization, X.Y.; Supervision, X.Y.; Project Administration, X.Y.; Funding Acquisition, X.Y. All authors have read and agreed to the published version of the manuscript.

**Funding:** This research was funded by China Institute of Rural Studies, Tsinghua University Doctoral Dissertation Scholarship, 2018–2019, grant number 201822.

**Acknowledgments:** The authors sincerely thanks researcher Mingli Wang of the Chinese Academy of Agricultural Sciences for his suggestions during the analysis, as well as Cha Cui, Pengcheng Zeng, Shubin Wang, Meng Zhang, Zengbo Bai and others for participating in the data collection for this paper. The authors thank the farmers in the study areas for their collaboration during interviews. The authors thank three peer anonymous reviewers' good suggestion to improve this paper.

**Conflicts of Interest:** The authors declare no conflict of interest.

## Appendix A

(Note: the full text currency conversion is 100 USD ($) = 698.23 CNY (¥), this data is from https://www.boc.cn/sourcedb/whpj/index_1.html 2020-02-05 14:18:42; in addition, 1 hectare = 15 mu. 10,000 CNY per mu (¥/mu) = 21,482.89 USD per ha ($/ha)

How do we understand the MBDC method?

Analysis of 18 price selection behaviors was based on a questionnaire using MBDC method, and it showed that 697 farmers responded to 18 prices differently. Farmers always make their mind based on the alternative flexibility between their LCRs and other items.

In order to understand the responses of different farmers to the designed 18 prices intuitively, Table A1 took four different types of farmers as examples. If the LCR price were between ¥10,000/mu ($21,480/ha) and ¥13,000/mu ($27,924/ha), farmer 1 would scale up; if the price were between ¥13,000/mu ($27,924/ha) and ¥20,000/mu ($42,960/ha), the farmer would downsize; if the price were over ¥30,000/mu ($64,440/ha), the farmer would definitely sell their LCRs. For farmers 3 and 4, because they did not rely on agriculture for a long time, no matter how low the price were, they would not buy the LCRs; but once the LCR price were up to ¥30,000/mu ($64,440/ha), they would sell it without hesitation. For farmers 5 and 6, the utility of their LCRs was relatively larger, so if the price were less than ¥10,000/mu ($21,480/ha), they would buy LCRs; however, no matter how high the price were, they would not sell the LCRs. Farmers 7 and 8 were too old to engage in agriculture, so they would sell rather than buy LCRs.

**Table A1.** Examples of 8 farmers' pricing selections for 18 prices.

| Pricing for LCR (¥10,000/mu)/($10,000/ha) | Farmer 1 | Farmer 2 | Farmer 3 | Farmer 4 | Farmer 5 | Farmer 6 | Farmer 7 | Farmer 8 |
|---|---|---|---|---|---|---|---|---|
| 0.3/0.64 | 5 | 5 | 3 | 3 | 4 | 5 | 3 | 3 |
| 0.4/0.86 | 5 | 5 | 3 | 3 | 4 | 5 | 3 | 3 |
| 0.5/1.07 | 5 | 5 | 3 | 3 | 4 | 5 | 3 | 3 |
| 0.6/1.29 | 5 | 5 | 3 | 3 | 4 | 5 | 3 | 3 |
| 0.8/1.72 | 5 | 5 | 3 | 3 | 4 | 5 | 3 | 3 |
| 1/2.15 | 4 | 5 | 3 | 3 | 4 | 3 | 2 | 3 |
| 1.3/2.79 | 3 | 5 | 3 | 3 | 3 | 3 | 2 | 3 |
| 2/4.30 | 2 | 5 | 3 | 3 | 3 | 3 | 1 | 3 |
| 3/6.44 | 1 | 4 | 2 | 1 | 3 | 3 | 1 | 3 |
| 4/8.59 | 1 | 2 | 2 | 1 | 3 | 3 | 1 | 1 |
| 5/10.74 | 1 | 1 | 1 | 1 | 3 | 3 | 1 | 1 |
| 6/12.89 | 1 | 1 | 1 | 1 | 3 | 3 | 1 | 1 |
| 7.5/16.11 | 1 | 1 | 1 | 1 | 3 | 3 | 1 | 1 |
| 10/21.48 | 1 | 1 | 1 | 1 | 3 | 3 | 1 | 1 |
| 12.5/26.85 | 1 | 1 | 1 | 1 | 3 | 3 | 1 | 1 |
| 15/32.22 | 1 | 1 | 1 | 1 | 3 | 3 | 1 | 1 |
| 20/42.96 | 1 | 1 | 1 | 1 | 3 | 3 | 1 | 1 |
| 30/64.44 | 1 | 1 | 1 | 1 | 3 | 3 | 1 | 1 |
| Min-WTS | 2.3 | 5 | 3.5 | 2 | —— | —— | 1 | 3.8 |
| Max-WTB | 1 | 3 | —— | —— | 1 | 0.8 | —— | —— |
| Reasons | Scale up | Grow vegetables | No longer want to farm | Not farming for a long time | Scale up | Scale up | Too old to farm | For pensions |

## Appendix B

Here is a real case of LCR transactions across collective boundaries.

It is legal to trade LCRs across collectives in Liangping District, Chongqing, one of the reform pilot areas. Mr. Shou Xiaojiang is a farmer in Renhe Village, Jindai Town, Liangping County. He used to be a soldier and was good at breeding catfish, but it was difficult for him to find a suitable site in his hometown collective because breeding catfish requires better environment. After a great effort, he found that a 15-mu (1-ha) river beach in Yihe Village, Jiaolong Town, was suitable for catfish farming. However, the risk of fishery investment was high, supporting and management houses were also insufficient, and farmers renting land to him might break the contract halfway, so a short-term contract was not feasible. Therefore, he wanted to obtain the long-term LCRs.

After several negotiations, with 20 farmers involved, the deal was finally concluded at a price of ¥34,500/mu ($74,106/ha). A total of 20 farmers sold their LCRs of 15 mu (1 ha) to the village collective. At the same time, Shou Xiaojiang gained membership in Yihe Village by paying the entrance fee of ¥3,000 ($429.6) in the form of "agricultural relocation" with the consent of Yihe village committee. Then, Shou Xiaojiang paid for the LCRs, and the village collective distributed the money to the farmers. At last, Shou Xiaojiang bought 50-year LCRs successfully.

All three parties legally benefited. First, Shou Xiaojiang no longer worried about investing in the site. Second, the 20 farmers thought it was cost-effective to take a compensation because the income from land was lower, and there were various problems in land transactions, such as the endless meetings, the difficulty in collecting rents, and the difficulty to get compensation through national land acquisition because the land is remote. Finally, the village collective charged a fee without changing land use.

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
