# Peer review of "The Potential Supply and Demand of Farmers’ Land Contract Rights-Based on 697 Households in Four Provinces of China"

_land, doi:10.3390/land9030080_

Round 1

Reviewer 1 Report

Summary

The article investigates the potential supply and demand of farmers’ land contract rights by applying a survey amongst farmers to get their opinion about the lowest price for selling or the highest price for buying land. A self-declared (SD) and a multiple bounded discrete choice (MBDC) were carried out to obtain reliable prices. In addition, the authors investigated the influence of factors for potential supply and for potential demand.

Overall comments:

The introduction leads in adequate manner to the topic. Literature cited is relevant and sufficiently supporting the research questions.

The applied methods in general are proper to answer the research question.

The outlined results are sound.

The weak point of the paper is the discussion part. It repeats the results of the previous chapter and the discussion is only based on “opinions of the authors”. Evidence about external justification of results (literature or interviews with other experts are missing).

Proposals for improvement - Overall comments

  • A map indicating the study area would be nice
  • Give a more detailed information about the procedure of “randomly selection”. What is the universal set? Why 740 questionnaires? Why did you vary the number of samples between the villages (10-20?). What was the random criteria for the samples in villages in town?
  • Did you outline a spatial analysis as you selected 18 villages. If not, justify it.
  • Reduce the number of decimal places to one to make text and tables easier to read (in case, where it is possible without losing significance)
  • Justify your “findings” in the discussion by external sources (literature, expert interviews)

Proposals for improvement - Detailed comments

33:     Describe shortly “the reform”

70:     $143,200 instead of $0.1432 million

161:   Add reference for Marshallian demand function

171:   Add reference for Miller’s thery

212ff: Add a table with figures aboutmaterial sources for better understanding

219:   you are referring to Ningxia proince – in the abstract you are mentioning also three other provinces ???

220:   Clarify the sentence “18 villages in 4 towns” as in my (European) understanding villages and towns are different administrative units at same level

303:   Reference [43] is identical with [5]

309:   Add number of reference in text [1]

312:   Add number of reference in text [2]

433ff: Revise discussion part and conclusions

Reviewer 2 Report

Dear Authors,

These are comments for the improvement of the manuscript.

  1. I suggest a careful check of English and the meaning of used words, in different paragraphs. Yet, this is not very critical

 Example:

Lines 63-64: LCRs has changed from  an emphasis on maintenance to an emphasis on transfer

Line 76:  due to funding restrictions ( meaning ?)

Lines 86:  potential suppliers and demanders  ( or buyers ??)

Lone 119:  condition of vague pricing ( meaning ?)

Line 212: primary materials : what does it mean? Data ?  or ……

Line 237: Reseach Method

Lines  458-459:  farmers who have settled in cities cannot 458 completely eliminate being land-bound  ( meaning ??)

  1. Theoretical mechanism analysis: This should be “analytical approach”

The functions and theories presented in this section are not clearly explained for the readers. It would be  good to capture the interests of the readers by making clearly what these theories and functions are about !!

I am not sure if you should say:  Households with Complete Elasticity of Substitution ( or increasing elasticity..)  the same for Zero or Decreasing??

When you say:   Households between Complete and Zero Elasticity of Substitution and thereafter

 close to complete or close to zero:  it is not between  !!

  1. Research Method

-The self-declared (SD) method, which is similar to open-ended questions: you should simply let the readers understand the methods your applied without introducing these complicated concepts

The same for contingent valuation (CV) method

Formulae and analytical methods presented in sections: 2 (2.1 and 2.2.) , 32.1; 32.2, 3.2.3; , 3.2.4.;  3.3.,  should be combined and summarized under one section, with sub-sections.  It is not clear how you are applying various analytical methods!

  1. Analytical variables:

Some variables need to be well discussed and presented: for example, what do you mean by land quality? indicators of land quality should have been used in the assessment: soil depth, fertility, slope, exposition to sun, climate conditions, topography, etc.

 The same question for the Risk appetite: what does it mean?? what are the realted indicatros? Clarify this in the text

Reviewer 3 Report

The title of paper is good but I think there is a space for better Title (this partof title is little bit confusing for potential readers:  "...Based on 697 Households in Four Provinces of China"

Define term WTS and WTB in rows 388 or 389

Please express all financial values in yuans and US dollars

Please express all area values in Chinese measure and also in hectares.

Round 2

Reviewer 1 Report

The article was improved according to my suggestions.

There are still two weakpoints to be corrected:

  • Figure 1 has to be improved: use the same grid Annotation for both Maps and indicate in the overall map the shape of the detailed Maps.
  •  Text has to corrected by an English native speaker.
